# FlashTP: Fused, Sparsity-Aware Tensor Product for Machine Learning Interatomic Potentials

**Seung Yul Lee**[1] **Hojoon Kim**[1] **Yutack Park**[1] **Dawoon Jeong**[1] **Seungwu Han**[1,2] **Yeonhong Park**[1] **Jae W. Lee**[1]

## Abstract

Machine Learning Interatomic Potentials (MLIPs) enable efficient molecular dynamics (MD) simulations with high accuracy. While equivariant MLIPs achieve state-of-the-art accuracy, they face significant computational bottlenecks centered around their Tensor-Product layer, which account for up to 75% of training time and cause substantial memory overhead. We present FlashTP, a highly optimized tensor-product library that addresses these inefficiencies through kernel fusion, sparse computation, and path-aggregated execution. FlashTP achieves up to $41.6\times$ and $60.8\times$ kernel speedups over *e3nn* and NVIDIA cuEquivariance, respectively. For SevenNet-l3i5, it delivers $4.2\times$ and $3.5\times$ speedup while reducing peak memory usage by $6.3\times$ and $6.2\times$ for inference and training, respectively. The code is available at https://github.com/SNU-ARC/flashTP.

## 1. Introduction

Machine learning interatomic potential (MLIP) is a method to predict energy and forces given atom positions and atomic numbers with machine learning models. It is gaining significant attention as a means to accelerate molecular dynamics (MD) simulations, which play a crucial role in computational materials science and chemistry. MLIP enables simulations to run orders of magnitude faster than quantum mechanical methods while preserving their accuracy (Unke et al., 2021; Deringer et al., 2021; Ko & Ong, 2023).

Among the various types of MLIPs, equivariant MLIPs (Thomas et al., 2018; Batzner et al., 2022; Batatia et al., 2022; Musaelian et al., 2023; Liao et al., 2024) have demonstrated superior performance across complex benchmarks that assess not only energy and

force errors but also real physical observables derived from MD simulations (Fu et al., 2022; Kim et al., 2023). Equivariant MLIPs typically follow the principles of graph neural networks, taking as input a graph where atoms are represented as nodes and their interactions as edges.

Equivariant MLIP models, however, suffer from slow inference and training speeds, as well as high memory requirements (Passaro & Zitnick, 2023; Luo & Krishnapriyan, 2024; Zhang et al., 2024; Xie et al., 2024). The key bottleneck of equivariant MLIP model lies in its Tensor-Product layers, which serves as the core operation responsible for updating the hidden states of nodes while incorporating interactions with neighboring nodes. For instance, in SevenNet-l3i5 (Park et al., 2024a;b), a state-of-the-art equivariant MLIP model, the Tensor-Product layer accounts for 89% of the inference time and 75% of training time. Moreover, the Tensor-Product layer operates on edges rather than nodes, leading to a colossal memory footprint since the number of edges far exceeds the number of nodes (e.g., $37\times$ more edges on average for SevenNet on MPF dataset (Chen & Ong, 2022)).

The inefficiency of the Tensor-Product layer stems from three sources. First, the Tensor-Product layer consists of multiple distinct kernels, and the intermediate data exchanged between these kernels generate substantial memory traffic. Second, the output data produced by the Tensor-Product layer is significantly large, leading to memory spikes that limit the scalability of MD simulations. Lastly, a large amount of ineffectual computation arises from the high sparsity of the Clebsch-Gordan (CG) coefficient matrix (Varshalovich et al., 1988), a constant matrix that ensures equivariance of updated node hidden states after their tensor product with connected edges and neighbors.

To address the aforementioned issues, we present FlashTP, a specialized Tensor-Product library designed to enhance the efficiency of equivariant MLIP model by both accelerating its speed and reducing memory requirements. FlashTP fully fuses all operations within the Tensor-Product layer, significantly reducing the memory traffic caused by intermediate data. Additionally, the fully fused Tensor-Product layer is again fused with its subsequent layer, eliminating memory spikes that would otherwise be caused by large output

[1]Seoul National University [2]Korea Institute for Advanced Study. Correspondence to: Jae W. Lee <jaewlee@snu.ac.kr>, Yeonhong Park <ilil96@snu.ac.kr>.

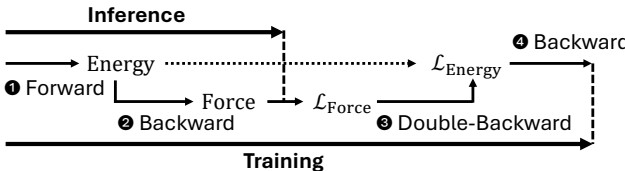

Figure 1. Overview of MLIP model inference and training pipeline. $\mathcal{L}_{\text{Force}}$ and $\mathcal{L}_{\text{Energy}}$ indicates the loss of the predicted forces and energy, respectively.

tensors. FlashTP also optimizes computation by skipping ineffectual operations. Lastly, as an additional optimization, FlashTP introduces path-aggregation, a technique that increases input reuse within the Tensor-Product layer, further accelerating the execution speed.

Through these optimizations, when applied to the state-of-the-art SevenNet-l3i5 model on MPF dataset, FlashTP achieves 4.2× faster inference and 3.5× faster training compared to *e3nn* (Geiger & Smidt, 2022; Geiger et al., 2022), a widely used framework for building equivariant MLIP models, while reducing peak memory usage by 6.3× and 6.2× for inference and training, respectively.

**Summary of Contributions.**

- We identify memory traffic, memory consumption, and ineffectual computations as the key bottlenecks in the Tensor-Product layer.

- We develop novel optimization techniques including kernel fusion, sparse computation skipping, and path-aggregated execution to address these bottlenecks.

- We demonstrate that FlashTP accelerates both inference (4.2×) and training (3.5×) for a state-of-the-art equivariant MLIP model.

- We show that FlashTP reduces peak memory usage by 6.3× for inference and 6.2× for training in equivariant MLIP model.

## 2. Background

### 2.1. Machine Learning Interatomic Potential (MLIP)

**MLIP Basics.** Energy and atomic forces estimated using interatomic potentials govern the reliability of molecular dynamics (MD) simulations, which are widely used in materials science and chemistry. Recently, machine learning interatomic potentials (MLIPs) have gained significant attention as a promising solution for these estimations. Trained on high-accuracy data generated using quantum mechanical methods, MLIP models predict energy, from which forces can be derived by computing gradients, achieving both high accuracy and computational efficiency. MLIP models have

proven to be effective in a wide range of applications, including the discovery of new materials (Hwang et al., 2023; Kruglov et al., 2023), simulations of semiconductor processes (Hong et al., 2024), and modeling of solid-state electrolytes (Wang et al., 2022; Lee et al., 2024) for safer and more energy-efficient batteries.

**MLIP Inference and Training Pipeline.** Figure 1 illustrates the inference and training pipeline of MLIP models. Inference of MLIP models consists of two phases. The first phase involves the forward computation of the model to predict the total energy (❶). The second phase follows, during which the forces are computed by taking the gradient of the energy (❷). This corresponds to the backward computation of the first phase. It is important to note that this backward computation is performed with respect to the result of the first phase (energy) and not its loss, making it distinct from backpropagation used for parameter updates.

For training, two additional phases are performed to update the model parameters. These correspond to the backward computations of the two inference phases. First, a backward-of-backward or double-backward computation is performed with respect to the loss of the predicted forces (❸). Following this, the backward computation is performed with respect to the loss of the predicted energy (❹). While phases ❷ and ❹ are computationally identical, they differ in their inputs: phase ❷ takes the energy as input, while phase ❹ takes the loss of energy as input.

### 2.2. Equivariant MLIP

Equivariant MLIPs (Batzner et al., 2022; Batatia et al., 2022; Musaelian et al., 2023; Passaro & Zitnick, 2023; Liao et al., 2024; Park et al., 2024b) are a class of interatomic potentials where both the predicted atomic forces and all intermediate layer outputs transform consistently with rotations of the molecular system (Thomas et al., 2018). These models have proven to be more data-efficient (Cohen & Welling, 2016; Batzner et al., 2022) and excel at predicting physical properties in MD simulations (Fu et al., 2022; Kim et al., 2023). While non-equivariant models exist (Schütt et al., 2017; Chen et al., 2019; Gasteiger et al., 2020; 2021), we focus on equivariant MLIPs due to their superior accuracy and widespread adoption in the field.

**Model Architecture.** Figure 2(a) illustrates the architecture of NequIP (Batzner et al., 2022), which serves as the foundation for typical equivariant MLIP models such as MACE (Batatia et al., 2022) and SevenNet (Park et al., 2024b). The model is based on the principles of graph neural networks (GNNs), taking a graph as input where nodes correspond to atoms and edges denote atomic interactions (Gilmer et al., 2017; Reiser et al., 2022). These models aim to generate latent representations for all nodes in the graph, from which each node's energy is predicted.

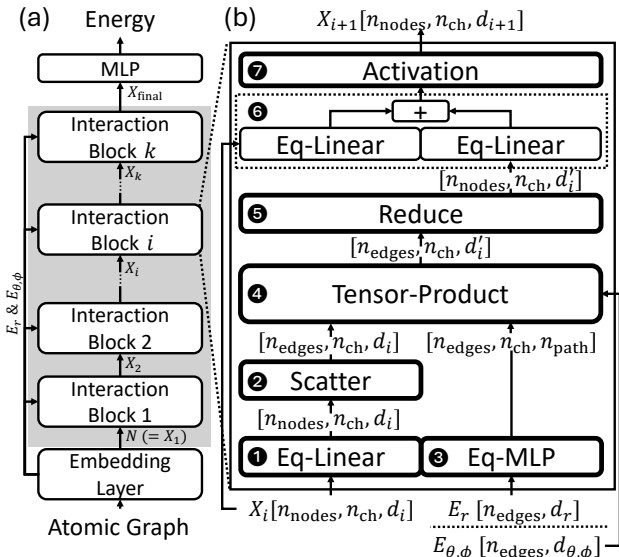

- $n_{\text{nodes/edges/ch/path}}$: number of nodes/edges/channels/paths
- $d_{r/\theta,\phi}$: dimension of radial/angular feature
- $d_i$: hidden dimension of $i$th layer
- $d'_i$: intermediate dimension of $i$th layer ($\gg d_i$)

*Figure 2.* Equivariant MLIP model architecture. Assumes all degrees of hidden feature to have same number of channel.

The model first passes through an embedding layer, where each node and edge of the graph are given their own features. Node features ($N$) are trainable parameters specific to each atomic number. Edge features consist of two components: one encoding distance information ($E_r$) and the other capturing angular relationships ($E_{\theta,\phi}$) between connected atoms. It then progresses through multiple interaction blocks that perform message passing, a mechanism for updating node representations by aggregating information from neighboring nodes (Gilmer et al., 2017). The first block receives the node features as hidden states, which are then iteratively updated across subsequent blocks ($N(= X_1) \to X_2 \to ... \to X_k$). Edge features are used in all blocks. The final hidden states ($X_{\text{final}}$) are used to predict energy, from which forces can be derived by computing its gradient.

Figure 2(b) illustrates the operations within each interaction block. The core of the interaction block is Tensor-Product layer (❹), where the message-passing mechanism is executed. Before reaching this layer, hidden states first pass through the equivariant linear layer (❶), which applies a linear transformation while preserving the input-output dimension. The hidden states then undergo a scatter layer which expands their first dimension from $n_{\text{nodes}}$ to $n_{\text{edges}}$ (❷). Meanwhile, $E_{\theta,\phi}$ is directly fed into Tensor-Product layer while $E_r$ passes through an MLP before reaching it (❸). After Tensor-Product layer, the expanded first dimen-

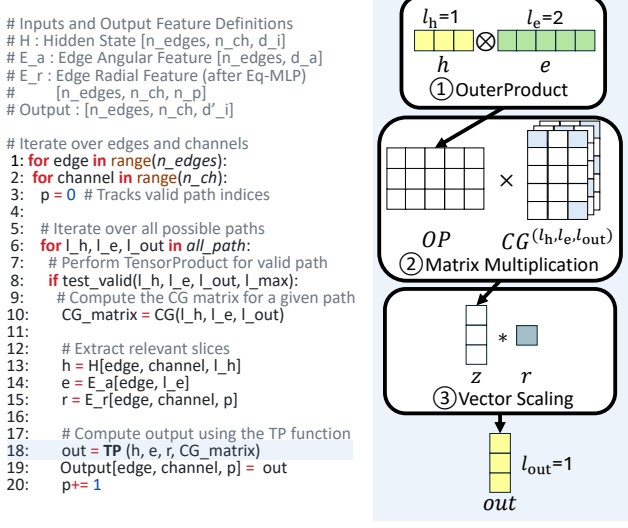

**(a)** Pseudocode of Tensor-Product Layer    **(b)** A single tensor-product

*Figure 3.* Overview of operation of Tensor-Product layer and a single tensor-product.

sion of the hidden states is reduced in a reduce layer (❺). Note that after passing through Tensor-Product layer, the hidden dimension increases from $d_i$ to $d'_i$, which is an order of magnitude larger. The hidden states then pass through another equivariant linear layer with a residual connection (❻), followed by an activation function (❼), producing the output of the block (or the input to the next block).

### 2.3. Tensor-Product Layer in Equivariant MLIP

**Operations in Tensor-Product Layer.** Figure 3(a) presents the pseudo-code for the Tensor-Product layer. The Tensor-Product layer performs a large number of tensor-products. Note that we distinguish the tensor-product (lowercase), which is an operation within the Tensor-Product layer, from the Tensor-Product layer itself (capitalized). Specifically, the tensor-product is carried out for each edge ($n_{\text{edges}}$), channel ($n_{\text{ch}}$) and path ($n_{\text{path}}$), resulting in a total of $n_{\text{edges}} \times n_{\text{ch}} \times n_{\text{path}}$ tensor-products (Line 1-6).

A path is defined by a combination of hidden, edge, and output degrees ($l_{\text{h}}, l_{\text{e}}, l_{\text{out}}$). Each of these degrees can take any value from 0 to $l_{\text{max}}$, meaning that the total possible number of paths is $(l_{\text{max}} + 1)^3$. $l_{\text{max}}$ is a user-defined parameter that controls the trade-off between model quality and computational cost. A larger $l_{\text{max}}$ improves accuracy (Batzner et al., 2022) but results in an exponential increase in computation due to the growing number of paths (Passaro & Zitnick, 2023; Luo & Krishnapriyan, 2024).

Meanwhile, when physical constraints are considered, some of $(l_{\text{max}} + 1)^3$ path are considered to be invalid, as the tensor-product of these paths always results in 0. Only paths

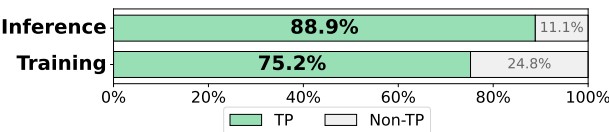

*Figure 4.* Portion of Tensor-Product layer in inference and training of SevenNet-l3i5.

*Table 1.* Sparsity in CG coefficient matrix for varying $l_{max}$

| $l_{max}$ | 1 | 2 | 3 | 4 | 5 |
|---|---|---|---|---|---|
| Sparsity (%) | 71 | 78 | 82 | 84 | 86 |

satisfying Inequality 1 produce non-zero output (Line 8).

$$|l_h - l_e| \leq l_{out} \leq l_h + l_e \qquad (1)$$

**Computations in Single Path** A tensor-product for a single path, defined by a 3-tuple $(l_h, l_e, l_{out})$ (Line 18), is illustrated in Figure 3(b). It receives a subvector of hidden states ($h$) and a subvector of angular features ($e$) as inputs, whose sizes are $(2 \times l_h + 1)$ and $(2 \times l_e + 1)$, respectively. It then produces an output ($out$) whose size is $(2 \times l_{out} + 1)$.

The computations proceed as follows: First, an outer product between $h$ and $e$ is performed, producing $OP$. This tensor is then multiplied by $CG^{l_h, l_e, l_{out}}$, a *Clebsch-Gordan* (CG) coefficient matrix (Varshalovich et al., 1988) corresponding to this path. CG coefficient matrix encapsulates the interaction principles between node and edge tensors preserving the equivariance of the output. Finally, the result of the matrix multiplication, denoted as $z$, is scaled by $r$, an element of the radial edge feature ($E_r$) after Eq-MLP, to produce the final output, $out$.

**Overhead of Tensor-Product Layer.** Due to its large amount of computations, which scale exponentially with $l_{max}$, the Tensor-Product layer represents the primary bottleneck in the equivariant MLIP model architecture. Figure 4 shows the breakdown of inference and training time for SevenNet-l3i5 (Park et al., 2024b) on the MPF dataset (Chen & Ong, 2022). The Tensor-Product layer accounts for approximately 89% and 75% of the inference and training time, respectively.

## 3. Inefficiency in Tensor-Product Layer

In this section, we highlight three major sources of inefficiencies in the Tensor-Product layer: 1) memory traffic from intermediate data, 2) peak memory spikes due to output data and 3) sparsity in the CG coefficient matrix.

### 3.1. Memory Traffic from Intermediate Data

A substantial amount of intermediate data is generated within the Tensor-Product layer. Figure 5 illustrates how memory accesses occur during the forward, backward, and double-backward phases for a single path of the tensor-

product, where the hidden degree dimension is 3, the edge degree dimension is 5, and the number of edges is 1K. Each phase consists of multiple separate kernels (3 for forward, 5 for backward, and 13 for double-backward), in which intermediate data are generated. Among these, the primary bottleneck is the outer product result generated during the forward phase (❶→❷), along with the data associated this outer product result in the backward phase (❷→❸/❹) and the double-backward phase (❶/❷→❸ and ❸→❹). These intermediate data, highlighted in red in Figure 5, scale linearly with the product of the node degree dimension and the edge degree dimension, making them significantly larger than the input degrees.

**Profiling Results.** Figure 6 illustrates the compute and DRAM bandwidth utilization during the forward, backward, and double-backward phases of a Tensor-Product layer of SevenNet-l3i5. The backward and double-backward phases take a significantly longer time than the forward phase, and they are largely memory-bound. This emphasizes the need to reduce memory bandwidth bottlenecks by minimizing intermediate data.

### 3.2. Peak Memory Spikes due to Output Data

In addition to the intermediate data, the output of the Tensor-Product layer presents a significant challenge, this time in terms of peak memory usage rather than memory bandwidth. This issue stems from the substantial increase in hidden dimension after passing through the Tensor-Product layer, growing from $d_i$ to $d_i'$. As mentioned earlier, $d_i'$ is usually an order of magnitude larger than $d_i$, making the output size $n_{edges} \times n_{ch} \times d_i'$ considerably large. Although the subsequent reduce layer immediately consumes and compresses this data, the full output must still be stored in memory beforehand, leading to memory spikes. In fact, this alone can increase peak memory consumption by a factor of four, severely limiting the number of atoms that can be included in simulations and posing a major scalability challenge.

### 3.3. Ineffectual Computation due to Sparsity

In the forward phase of the Tensor-Product layer, the outer product of input degrees (hidden degree and edge degree) is multiplied by the CG coefficient matrix (❷ in Figure 5(a)). Meanwhile, this matrix is highly sparse, with a sparsity ranging from approximately 71% to 86%, as shown in Table 1. This implies that the matrix multiplication between the outer product result and the CG coefficient matrix involves a significant amount of ineffectual computation. These inefficien-

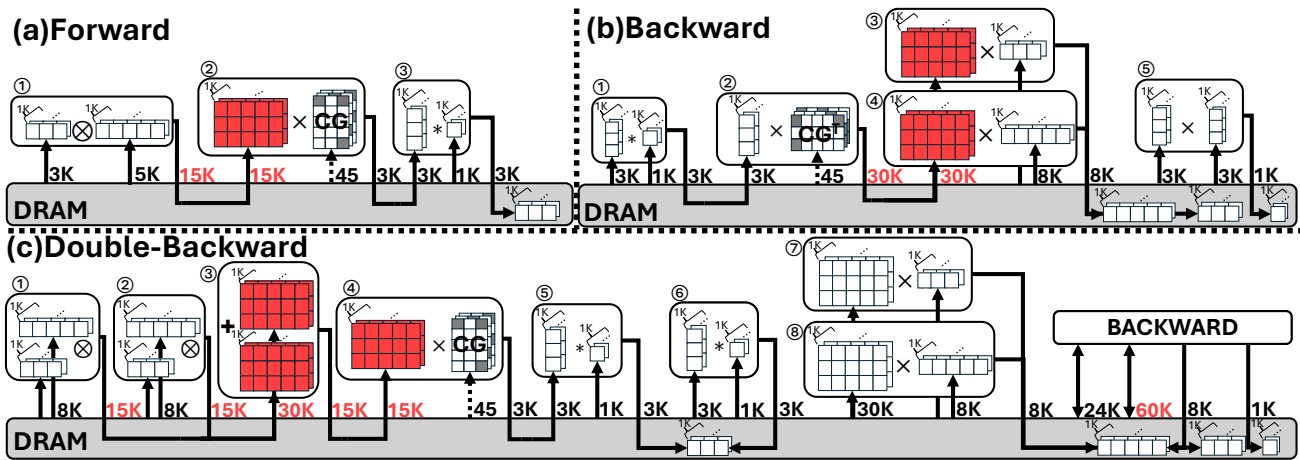

*Figure 5.* Kernels for the three phases (forward, backward, double-backward) of a tensor-product. Arrows represent memory traffic, with the numbers indicating the amount of traffic (in elements), calculated for the tensor-product with 1K edges. The *Clebsch-Gordan* (CG) coefficient matrix are reused across all edges, contributing only 45 elements to the total memory traffic. Elements placed inside the DRAM indicates the output of the tensor-product.

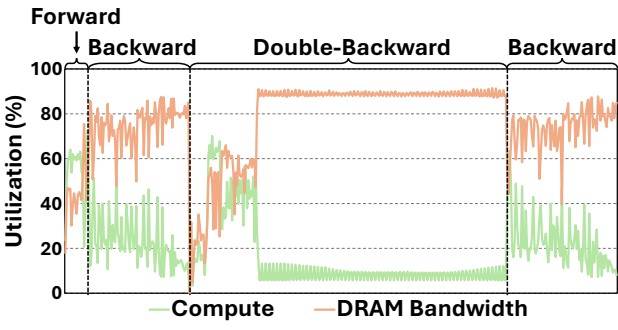

*Figure 6.* Compute and DRAM bandwidth utilization of a Tensor-Product layer in SevenNet-l3i5 on an NVIDIA A100 GPU. Time regions of forward, backward and double-backward are labeled.

cies propagate also to the backward and double-backward phases, particularly in the backward and double-backward kernels corresponding to this matrix multiplication (❸/❹ in Figure 5(b) and ❹ in Figure 5(c)).

# 4. FlashTP

## 4.1. Overview

Addressing the inefficiencies identified in Section 3, we introduce FlashTP, a tensor-product GPU library for equivariant MLIP. To mitigate the first two inefficiencies—intermediate data and output data—FlashTP fully fuses all kernels within the Tensor-Product layer, as well as its subsequent layer, into a single kernel (Section 4.2). Additionally, it exploits the sparsity of the CG coefficient matrix

to eliminate redundant computations, addressing the third inefficiency (Section 4.3). Finally, FlashTP introduces a technique to reduce memory traffic associated with tensor-product input data, which emerges as a new bottleneck after resolving the three aforementioned issues (Section 4.4).

## 4.2. Kernel Fusion

Kernel fusion is a widely used technique that combines multiple kernels into a single kernel to eliminate data movement between them (Dao et al., 2022). FlashTP employs kernel fusion for the Tensor-Product layer at two levels: intra-layer kernel fusion and inter-layer kernel fusion. The former addresses overhead from intermediate data, while the latter addresses overhead from output data.

**Intra-Layer Kernel Fusion.** A key strategy for fusing kernels in the Tensor-Product layer—which involves executing many computation kernels—is to divide them into two phases: those up to and including multiplication by the CG coefficient matrix, and those that follow.

In the first phase, FlashTP fuses kernels by handling all operations associated with each non-zero element in the CG coefficient matrix in a single pass. For instance, for the entry at index $(1, 1, 0)$, it multiplies the $h[1]$ by the $e[1]$ and then by that CG coefficient. Because each pass yields only a partial sum, any operations depending on those CG coefficient matrix products must wait until the partial sums are fully accumulated. Once accumulation is complete, the second phase of fused computations can proceed. This two-phase approach eliminates redundant work and streamlines fusion in the later phase.

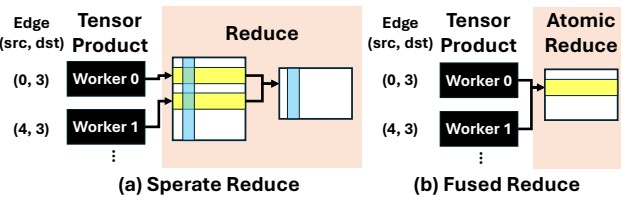

**(a) Sperate Reduce**      **(b) Fused Reduce**

*Figure 7.* Comparison of the output stage of the Tensor-Product layer (a) before and (b) after inter-layer kernel fusion, assuming two worker outputs share the same destination node (`dst`). In (a), each output is stored in memory before reduction. In (b), Each output is directly accumulated using an atomic reduction.

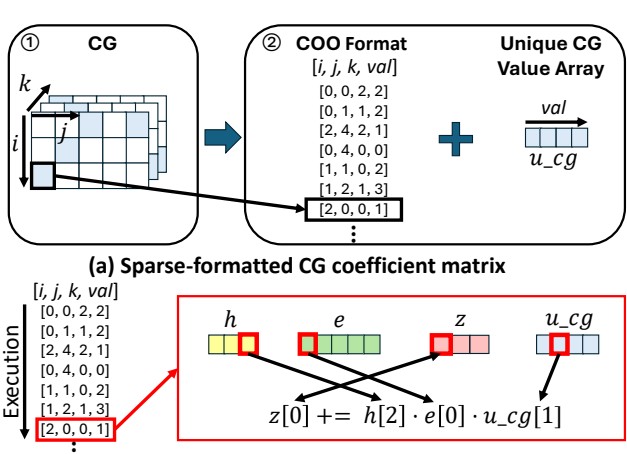

**(a) Sparse-formatted CG coefficient matrix**

**(b) Computation with sparse-formatted CG coefficient matrix**

*Figure 8.* Implementation of sparse tensor-product in FlashTP.

**Inter-Layer Kernel Fusion.** Figure 7 illustrates how the Tensor-Product layer is fused with subsequent layer, the reduce layer. The reduce layer accumulates tensor-product results for edges that share the same destination nodes. Since tensor-products for different edges are likely to be executed by different workers on GPUs (e.g., thread blocks in CUDA), synchronization issues may arise during fusion when multiple edges map to the same destination node. To address this, FlashTP employs atomic add operations in such cases to ensure correct reduction.

### 4.3. Applying Sparsity in Tensor-Product

In order to exploit the sparsity of CG coefficient matrix, FlashTP stores CG coefficient matrix in sparse data format. Figure 8(a) shows sparse data format used in FlashTP. The coordinate format (COO) is employed, where each nonzero value is represented as a 4-tuple: three indices corresponding to each dimension $(i, j, k)$ and the index of non-zero value $(val)$. To further enhance storage efficiency, FlashTP exploits the fact that the CG coefficient matrix contains many duplicated values. Specifically, FlashTP creates an

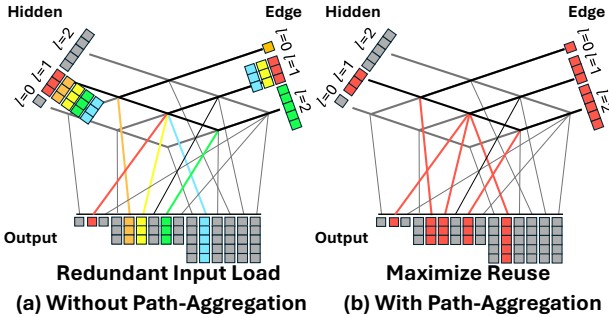

**Redundant Input Load**    **Maximize Reuse**
**(a) Without Path-Aggregation**    **(b) With Path-Aggregation**

*Figure 9.* Visualization of the effect of path-aggregation.

array of unique CG coefficient values ($u\_cg$) and stores only the index of each value within this array in the $val$ field of the tuple instead of the original value. This index can be represented with fewer bits (e.g., 8 bits) rather than storing the original 32-bit value. When performing computations with this sparse-formatted CG coefficient matrix, FlashTP iteratively processes each tuple one by one, as illustrated in Figure 8(b).

### 4.4. Path-Aggregation

FlashTP introduces path-aggregation, a technique designed to mitigate the new bottleneck—input load traffic in tensor products—that arises after the integration of kernel fusion and sparse computation. By grouping tensor-product paths that share the same input and executing them within a single kernel, this method reduces memory traffic.

Figure 9 illustrates the path-aggregation by visualizing connection between inputs and outputs involved in each path. Without path-aggregation, multiple paths may redundantly read the same input data (Figure 9(a)). For example, the degree-1 subvector of hidden states is read five times, and the degree-1 subvector of edge features is read three times. On the other hand, with path-aggregation, hidden states and edge features are read from memory only once, reducing memory traffic by factors of five and three, respectively, in this example (Figure 9(b)). In FlashTP, paths are grouped by the hidden subvector they use, as shown in Figure 9(b).

## 5. Implementation

**Programming Interface.** FlashTP provides programming interface that is easily compatible with *e3nn* (Geiger & Smidt, 2022; Geiger et al., 2022), a PyTorch-based framework for equivariant operations which is widely used for building equivariant MLIP models. Specifically, the interface of FlashTP well aligns with the implementation of TensorProduct class in *e3nn* as can be seen in Figure 10. This allows programmers to integrate FlashTP with a minimal

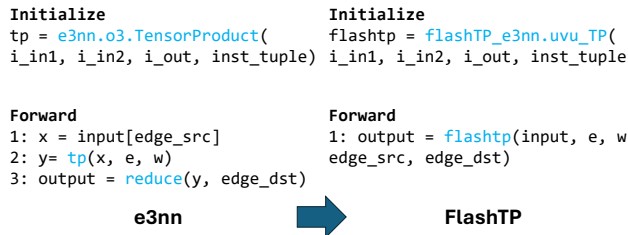

*Figure 10.* Code change necessary to integrate FlashTP into the existing implementation based on the *e3nn* framework.

code change. In addition, FlashTP preserves the same matrix storage ordering and model structure as *e3nn*, ensuring consistency and ease of adoption.

**Preprocessing.** Before performing inference or training, during model initialization, FlashTP requires preprocessing for two purposes. First, it identifies paths to aggregate by examining model configurations. Second, it constructs the CG coefficient matrix in a sparse format.

**GPU Parallelization Scheme.** The tensor-product operation in FlashTP is parallelized across three dimensions: the number of edges, the number of aggregated paths, and the channel dimension associated with the node degree for each path. For a single warp, FlashTP assigns tensor-product computations with the same path but of different channel and edge index.

This ensures that all threads access the same CG coefficient matrix metadata simultaneously allowing FlashTP to fully leverage constant memory for storing the metadata for sparse CG coefficient matrix. However, if multiple paths execute on a single compute unit using different CG coefficient matrices, cache evictions can occur in constant memory, leading to long cache miss stalls. FlashTP prevents this by assigns a single path to each thread block and maximizes its size. FlashTP automatically selects the block size by maximizing it while ensuring efficient shared memory utilization, thereby improving GPU occupancy.

## 6. Evaluation

We evaluate FlashTP against two comparison baselines: *e3nn* (Geiger et al., 2022) and cuEquivariance (cuEq) (Geiger et al., 2024). *e3nn* is a widely used framework for building equivariant MLIP models, while cuEq is a recently released CUDA library by NVIDIA that accelerates equivariant operations, such as tensor products. We first evaluate how much FlashTP accelerates each tensor-product operations (Section 6.1) and present how it translates into end-to-end speedup (Section 6.2). All evaluations, except for multi-GPU training, are conducted on a single NVIDIA A100 80GB GPU. Further details on the evaluation setup are provided in Appendix A.

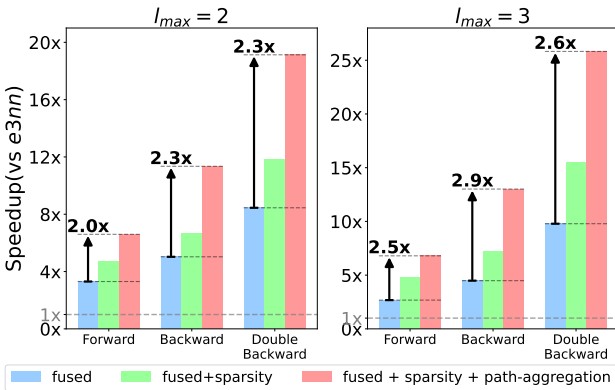

*Figure 11.* Ablation study result of three optimization techniques introduced by FlashTP. The speedup over *e3nn* is reported.

### 6.1. Kernel Microbenchmark

Table 2 presents latency for three phase (forward, backward, and double-backward) of tensor-product across varying $l_{max}$ configuration. For both single and double precision, FlashTP significantly outperforms *e3nn* for all cases. On average, for **single precision**, FlashTP achieves **6.75×**, **14.37×**, and **26.54×** speedups for the forward, backward, and double-backward phase, respectively. Similarly, for **double precision** (fp64), it achieves **6.38×**, **16.26×**, and **25.98×**, respectively. The speedup is particularly notable in the backward and double-backward phases, as they benefit the most from reduced memory traffic due to kernel fusion and path aggregation, given their highly memory-bound nature.

Compared to cuEq, FlashTP is also consistently faster in all cases. While cuEq demonstrates impressive results (e.g., cases 1 and 2) and occasionally comes close to FlashTP in performance, its efficiency diminishes as $l_{max}$ increases, limiting its applicability to high-precision MD simulations. This is because cuEq's efficiency is primarily attributed to its effective use of shared memory, which becomes infeasible for large $l_{max}$ due to its substantial memory footprint.

**Ablation Study.** Figure 11 presents how kernel performance improves with the introduction of each of the three optimization techniques of FlashTP: 1) kernel fusion, 2) sparse computation and 3) path-aggregation. For this evaluation, we use the case of $l_{max} = 2$ and $l_{max} = 3$. Results for other $l_{max}$ values are provided in Appendix A.3. All three optimization techniques significantly enhance kernel performance.

**Numerical Stability.** Since kernel fusion can alter the order of floating-point operations, it may produce slightly different results compared to unfused kernels, especially

*Table 2.* Speedup comparison of different tensor-product configurations relative to *e3nn* baseline for both single (fp32) and double precision (fp64), measured across varying $l_{\max}$. Entries marked with * indicate speedup relative to cuEq due to *e3nn* encountering out-of-memory (OOM) errors.

| | | Forward | | | Backward | | | Double-Backward | | |
|---|---|---|---|---|---|---|---|---|---|---|
| | | Latency (*ms*) (Speedup over *e3nn*) | | | Latency (*ms*) (Speedup over *e3nn*) | | | Latency (*ms*) (Speedup over *e3nn*) | | |
| | $l_{\max}$ | *e3nn* | cuEq | FlashTP | *e3nn* | cuEq | FlashTP | *e3nn* | cuEq | FlashTP |
| fp32 | 1 | 2.05 | 0.61 (3.4x) | **0.22 (9.4x)** | 6.18 | 0.60 (10.3x) | **0.54 (11.4x)** | 9.13 | 0.92 (9.9x) | **0.83 (11.0x)** |
| | 2 | 6.20 | 1.24 (5.0x) | **0.81 (7.7x)** | 32.26 | 2.28 (14.2x) | **1.86 (17.3x)** | 63.86 | 4.28 (14.9x) | **2.50 (25.6x)** |
| | 3 | 18.40 | 16.06 (1.1x) | **2.54 (7.3x)** | 85.00 | 91.68 (0.9x) | **6.15 (13.8x)** | 254.13 | 204.07 (1.2x) | **8.73 (29.1x)** |
| | 4 | 45.58 | 122.20 (0.4x) | **9.29 (4.9x)** | 268.48 | 835.63 (0.3x) | **17.89 (15.0x)** | 1196.89 | 1822.67 (0.7x) | **29.99 (39.9x)** |
| | 5 | 99.01 | 121.02 (0.8x) | **18.22 (5.4x)** | 779.14 | 858.95 (0.9x) | **51.96 (15.0x)** | 3752.63 | 1835.29 (2.0x) | **93.11 (40.3x)** |
| | | Forward | | | Backward | | | Double-Backward | | |
| | | Latency (*ms*) (Speedup over *e3nn*) | | | Latency (*ms*) (Speedup over *e3nn*) | | | Latency (*ms*) (Speedup over *e3nn*) | | |
| | $l_{\max}$ | *e3nn* | cuEq | FlashTP | *e3nn* | cuEq | FlashTP | *e3nn* | cuEq | FlashTP |
| fp64 | 1 | 2.39 | 0.76 (3.1x) | **0.44 (5.4x)** | 12.21 | 0.94 (13.0x) | **0.74 (16.4x)** | 17.96 | 1.55 (11.6x) | **1.19 (15.1x)** |
| | 2 | 9.50 | 2.13 (4.5x) | **1.44 (6.6x)** | 43.63 | 3.18 (13.7x) | **2.86 (15.3x)** | 95.40 | 6.66 (14.3x) | **4.36 (21.9x)** |
| | 3 | 30.10 | 41.38 (0.7x) | **4.00 (7.5x)** | 178.75 | 243.59 (0.7x) | **10.81 (16.5x)** | 594.36 | 532.85 (1.1x) | **17.97 (33.1x)** |
| | 4 | 74.33 | 56.33 (1.3x) | **11.93 (6.2x)** | 577.02 | 441.80 (1.3x) | **34.04 (16.9x)** | 2566.10 | 1158.80 (2.2x) | **61.72 (41.6x)** |
| | 5 | N/A | 134.47 | **30.18 (4.5x)*** | N/A | 1103.31 | **114.85 (9.6x)*** | N/A | 2861.31 | **199.18 (14.4x)*** |

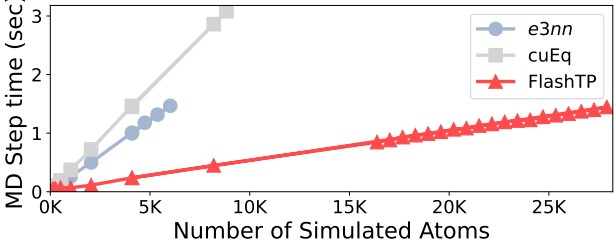

*Figure 12.* Comparison of average MD simulation step time across varying numbers of copper atoms, scaling up until simulations encounter an Out-of-Memory (OOM) error.

*Table 3.* Per-epoch training time of SevenNet models on a GPU

| | SevenNet-l2i5 | | SevenNet-l3i5 | | SevenNet-l4i5 | |
|---|---|---|---|---|---|---|
| | Time (*min*) | Peak Mem. (*GB*) | Time (*min*) | Peak Mem. (*GB*) | Time (*min*) | Peak Mem. (*GB*) |
| *e3nn* | 31 | 3.78 | 76 | 8.52 | 213 | 17.43 |
| cuEq | 23 | 2.56 | 97 | 5.23 | 358 | 9.97 |
| FlashTP | 20 | 0.89 | 22 | 1.37 | 32 | 2.07 |

for single-precision case. To quantify this potential error, we assume the fp64 results from *e3nn* as the golden reference and compare the deviations of *e3nn* fp32 results and FlashTP fp32 results from it. Across the forward, backward, and double-backward phases, the average root mean squared error for *e3nn* fp32 and FlashTP fp32 are comparable ($5.64 \times 10^{-6}$ vs. $4.60 \times 10^{-6}$), indicating that kernel fusion does not appear to have an adverse impact on numerical stability. Detailed results are in Appendix B.

### 6.2. End-to-end Speedup

**Inference.** Figure 12 presents the average time required for a single step of an MD simulation, which consists of a single model inference and an atomic position update, using pre-trained SevenNet-l3i5 model. We evaluate FlashTP alongside comparison baselines, increasing the number of atoms until an out-of-memory error occurs. Notably, FlashTP supports up to 28K atoms, 4.7× and 3.1× more than the limitations of *e3nn* and cuEq, respectively. This improvement is primarily attributed to FlashTP's kernel fusion, which eliminates memory spikes caused by the output data

of the Tensor-Product layer. In terms of simulation time, FlashTP also outperforms the comparison baselines. For a system with 4K atoms, FlashTP achieves a 4.2× speedup over *e3nn* and a 6.2× speedup over cuEq, while using 6.3× and 4.3× less memory compared to *e3nn* and cuEq, respectively.

**Single GPU Training.** Table 3 presents the training time per-epoch and peak memory usage for three variants of SevenNet: l2i5, l3i5, and l4i5, each corresponding to $l_{\max}$ values of 2, 3, and 4, respectively. FlashTP achieves speedups of 1.6×, 3.5×, and 6.7× over *e3nn* for the l2i5, l3i5, and l4i5 configurations, respectively. Notably, as the proportion of computation spent on the Tensor-Product layer increases with $l_{\max}$, the benefits of FlashTP become more pronounced. Compared to cuEq, FlashTP is 1.15×, 4.41×, and 11.19× faster. Consistent with the observations in Section 6.1, cuEq demonstrates impressive performance for $l_{\max} = 2$ but falls significantly behind for larger $l_{\max}$ values. In terms of peak memory usage, FlashTP requires significantly less memory than both comparison baselines, further emphasizing its efficiency in handling large systems.

**Multi-GPU Training.** Table 4 shows the per-epoch training time of the SevenNet-l3i5 model on a multi-GPU system. Each GPU node consists of eight NVIDIA A100 80GB GPUs, interconnected via NVLink and NVSwitch, with inter-node communication handled over a 100GB/s network.

*Table 4.* Per-epoch training time of SevenNet-l3i5 on multi-GPUs

| # of Nodes (# of GPUs) | 1 (8) | 2 (16) | 4 (32) | 8 (64) |
|---|---|---|---|---|
| *e3nn* (sec) | 735 | 399 | 218 | 123 |
| FlashTP (sec) | 164 | 87 | 49 | 30 |
| Speedup | 4.5× | 4.6× | 4.4× | 4.1× |

FlashTP demonstrates consistent speedup with an increasing number of GPUs. The slight reduction in speedup at higher GPU counts is likely due to inter-node communication overhead, which, in turn, reduces the effective region of interest for FlashTP —i.e., the portion of the workload that benefits from its acceleration.

## 7. Related Work

**Acceleration of Clebsch-Gordan Tensor Products.** Several approaches have been proposed to accelerate the Clebsch-Gordan Tensor Product (CGTP). One notable approach is the SO(2) Tensor-Product (Passaro & Zitnick, 2023), which reduces the number of non-zero elements in the Clebsch-Gordan coefficient matrix by aligning the axis via rotation, thereby lowering the computational complexity of CGTP. The SO(2) method is orthogonal to FlashTP and can be complementary, as FlashTP also benefits from a reduced number of non-zero elements in the Clebsch-Gordan coefficient matrix. Another approach, Fused Tensor Product (FTP) or `FusedTensor` (Unke & Maennel, 2024), accelerates CGTP by fusing all irreducible representations (irreps) into a single tensor that can be processed using standard matrix multiplication. The output is then decomposed to recover the original irreps. While this method reduces computational cost compared to traditional CGTPs, it comes at the expense of reduced expressivity (Xie et al., 2024).

**Alternatives to Clebsch-Gordan Tensor Products.** Alternative approaches have explored using different sets of bases beyond spherical harmonics to accelerate tensor-product computations. The Gaunt Tensor Product (GTP) (Luo & Krishnapriyan, 2024) changes the basis to the frequency domain and uses Fast Fourier Transforms to speed up tensor-product computations. While computationally efficient, this approach trades off some expressivity compared to CGTP. Additionally, due to the symmetric nature of its operations, GTP is unable to capture chiral features in 3D structures (Xie et al., 2024). Another notable direction involves Cartesian-based models, which use Irreducible Cartesian Tensors (ICT) instead of spherical harmonics. Recent studies have demonstrated promising performance in both accuracy and latency, particularly at lower ranks, when applied to MLIP tasks (Simeon & De Fabritiis, 2023; Zaverkin et al., 2024). Nonetheless, CGTP-based models remain the dominant paradigm for top-performing entries, as widely used MLIP benchmarks such as OC20 (Chanussot et al.,

2021) and Matbench (Riebesell et al., 2024) continue to rely on CGTP, underscoring the critical need for further acceleration of CGTP.

## 8. Conclusion

In this paper, we presented FlashTP, a solution to two key inefficiencies in the Tensor-Product layer of equivariant MLIP model models: the memory footprint from intermediate data and the sparsity of the CG coefficient matrix. Through kernel fusion, sparsity utilization, and path-aggregated execution, FlashTP achieved significant speedups in both inference and training of SevenNet-l3i5, a state-of-the-art equivariant MLIP model.

## Acknowledgments

This work was supported by a research grant from Samsung Advanced Institute of Technology (SAIT) and the National Research Foundation of Korea (NRF) grant funded by the Korean Government (MSIT) (RS-2024-00340008). We thank Saerom Choi and Yongdeok Kim from the Materials AI Lab at Samsung AI Center for providing GPU resources and supporting the multi-GPU evaluation of FlashTP on their server infrastructure.

## Impact Statement

This paper presents work whose goal is to advance the field of Machine Learning. There are many potential societal consequences of our work, none which we feel must be specifically highlighted here.

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

# A. Evaluation Details

We used pytorch 2.5.1+cu12.4, *e3nn* 0.5.4 (Geiger et al., 2022), cuEquivariance 0.2.0 (Geiger et al., 2024), Atomic Simulation Environment (ASE) 3.24.0 (Larsen et al., 2017) for evaluation.

GPU kernel runtime and memory usage measurements are conducted using the PyTorch profiler, leveraging its capability to track forward and backward connections for precise measurement of backward and double-backward operations. All evaluations, except for multi-GPU training, are performed on an NVIDIA A100 GPU with 80GB of memory. For multi-GPU training, GPU nodes that consist of eight NVIDIA A100 GPUs (80GB), interconnected via NVLink and NVSwitch, with inter-node communication handled over a 100GB/s network was used.

## A.1. Kernel Microbenchmark

We benchmark the Tensor-Product layer 4 from NequIP (Batzner et al., 2022) across different values of $l_{\max}$, keeping the channel of hidden feature fixed to 32 channels.

Our synthetic input graph has 512 nodes, each with 64 outgoing edges (32K edges in total). To generate connectivity, we assign each node 64 distinct random destinations drawn uniformly from the other 511 nodes. This exact graph topology is reused for all kernel implementations (*e3nn*, cuEq and FlashTP) to ensure fair comparison. All hidden, edge and weight values are randomly generated.

*Table 5.* Tensor-Product configurations for kernel microbenchmark

| $l_{max}$ | Hidden | Edge | Output |
|---|---|---|---|
| 1 | 32x0e+32x0o+32x1e+32x1o | 1x0e+1x1o | 64x0o+64x0e+96x1o+96x1e |
| 2 | 32x0e+32x0o+32x1e
+32x2e+32x1o+32x2o | 1x0e+1x1o+1x2e | 96x0o+96x0e+192x1o
+192x1e+192x2o+192x2e |
| 3 | 32x0e+32x0o+32x1e+32x2e
+32x3e+32x1o+32x2o+32x3o | 1x0e+1x1o+1x2e+1x3o | 128x0o+128x0e+288x1o+288x1e
+352x2o+352x2e+320x3o+320x3e |
| 4 | 32x0e+32x0o+32x1e+32x2e+32x3e
+32x4e+32x1o+32x2o+32x3o+32x4o | 1x0e+1x1o+1x2e+1x3o+1x4e | 160x0o+160x0e+384x1o+384x1e+512x2o
+512x2e+544x3o+544x3e+480x4o+480x4e |
| 5 | 32x0e+32x0o+32x1e+32x2e+32x3e+32x4e
+32x5e+32x1o+32x2o+32x3o+32x4o+32x5o | 1x0e+1x1o+1x2e+1x3o+1x4e+1x5o | 192x0o+192x0e+480x1o+480x1e+672x2o+672x2e
+768x3o+768x3e+768x4o+768x4e+672x5o+672x5e |

## A.2. Channel Scaling

We repeated the kernel microbenchmark for $l_{\max} = 3$ using hidden-channel dimensions of 64 and 128, reporting kernel runtimes in milliseconds. Across all channel sizes, FlashTP shows consistent speedup.

*Table 6.* Kernel microbenchmark results for $l_{\max} = 3$ across different channel sizes

|  | Forward | | | Backward | | | Double-Backward | | |
|---|---|---|---|---|---|---|---|---|---|
| **Channel Size** | **32** | **64** | **128** | **32** | **64** | **128** | **32** | **64** | **128** |
| *e3nn* | 18.4 | 34.25 | 66.8 | 85 | 163.96 | 324.73 | 254.13 | 503.28 | 1003.95 |
| FlashTP | 2.54 | 5.02 | 9.98 | 6.15 | 12.14 | 24.19 | 8.73 | 17.41 | 34.81 |
| Speedup | 7.2x | 6.8x | 6.7x | 13.8x | 13.5x | 13.4x | 29.1x | 28.9x | 28.8x |

## A.3. Ablation Study

Table 7 reports the ablation results on our kernel microbenchmark, showing the speedup over *e3nn* for each of the three optimization strategies—kernel fusion (Fused), sparsity optimization (Sparse), and path aggregation (Path)—as well as their combined effect (All).

Because the Clebsch–Gordan (CG) coefficient matrix for the non-sparse variants exceeds the capacity of constant memory, we disabled constant-memory storage for all ablation configurations, including the Sparse variant, to ensure a fair comparison. Consequently, the cumulative "All" configuration in the ablation study achieves a slightly lower speedup than the full FlashTP implementation.

*Table 7.* Ablation results on the kernel microbenchmark

| $l_{max}$ | Forward | | | | | Backward | | | | | Double-Backward | | | | |
|---|---|---|---|---|---|---|---|---|---|---|---|---|---|---|---|
| | e3nn | Fused | Path | Sparse | All | e3nn | Fused | Path | Sparse | All | e3nn | Fused | Path | Sparse | All |
| 1 | 1× | 3.46× | 4.37× | 4.46× | 5.25× | 1× | 10.95× | 15.85× | 12.16× | 18.41× | 1× | 9.70× | 12.80× | 10.87× | 14.77× |
| 2 | 1× | 3.30× | 4.16× | 4.68× | 6.60× | 1× | 5.03× | 7.26× | 6.63× | 11.35× | 1× | 8.45× | 11.85× | 11.86× | 19.13× |
| 3 | 1× | 2.67× | 3.22× | 4.79× | 6.80× | 1× | 4.48× | 6.32× | 7.24× | 13.01× | 1× | 9.79× | 12.45× | 15.46× | 25.82× |
| 4 | 1× | 2.32× | 2.47× | 4.51× | 6.02× | 1× | 3.97× | 4.99× | 7.41× | 13.16× | 1× | 11.56× | 14.58× | 19.89× | 30.73× |

## A.4. Roofline Analysis

We estimate the theoretical performance limits of FlashTP using a roofline analysis, based on the peak capabilities of the A100 GPU (19.5 TFLOPS and 1.9 TB/s memory bandwidth). According to this analysis, the estimated latencies for the fp32 kernel microbenchmark with $l_{max} = 3$ are 0.92 ms for the forward pass, 1.78 ms for the backward pass, and 2.70 ms for the double-backward pass. The gap between these theoretical bounds and our measured performance in Table 2 (Section 6.1) indicates that FlashTP has room for further optimization.

## A.5. End-to-end Inference

We evaluate end-to-end inference performance by integrating a pre-trained SevenNet-l3i5 model (Park et al., 2024a) into the ASE framework to run molecular-dynamics (MD) simulations of copper atoms. To gauge memory demands, we also determine the maximum number of atoms supportable, which correlates directly with peak memory usage. Each simulation spans 500 timesteps, and our reported runtimes are averaged over the final 100 timesteps for stability.

In our experiments, the out-of-memory (OOM) failures we observed when using FlashTP originate not from the Tensor-Product layer but from the default *e3nn* implementation of the Linear layer. This suggests that by optimizing the Linear layer implementation, FlashTP could scale to even larger atomic systems, highlighting its robustness and scalability for large-scale MD simulations.

## A.6. End-to-end Training

The models SevenNet-l2i5 (which is SevenNet-0) (Park et al., 2024b), SevenNet-l3i5, and SevenNet-l4i5 are trained using the same training configuration. The training was conducted using the MPF dataset (Chen & Ong, 2022), which comprises 168,921 samples, with each sample containing an average of 29 atoms. Training time is reported as the duration of a single epoch, and peak-memory reduction ratios are computed per batch and then averaged across multiple batches. For multi-GPU experiments, only SevenNet-l3i5 was used, maintaining a per-GPU batch size of 16 to match the single-GPU setup.

*Table 8.* Training configuration

| Training Parameters | | Model Parameters | |
|---|---|---|---|
| **Dataset** | MPF | **Precision** | FP32 |
| **Batch Size** | 16 | **Cutoff** | 6 |
| **Optimizer** | Adam | **Cutoff Function** | XPLOR (5.5) |
| **Learning Rate** | 0.001 | **Radial Basis** | Bessel (8 basis) |
| **Loss Function** | Huber Loss | **Activation** | SiLU (e), tanh (o) |
| **Scheduler** | Linear | | |

*Table 9.* SevenNet-l2i5 Tensor-Product layer configuration

| | SevenNet-l2i5 | | |
|---|---|---|---|
| | **Hidden** | **Edge** | **Output** |
| **Layer 1** | 128x0e | 1x0e+1x1e+1x2e | 128x0e+128x1e+128x2e |
| **Layer 2** | 128x0e+64x1e+32x2e | 1x0e+1x1e+1x2e | 224x0e+384x1e+352x2e |
| **Layer 3** | 128x0e+64x1e+32x2e | 1x0e+1x1e+1x2e | 224x0e+384x1e+352x2e |
| **Layer 4** | 128x0e+64x1e+32x2e | 1x0e+1x1e+1x2e | 224x0e+384x1e+352x2e |
| **Layer 5** | 128x0e+64x1e+32x2e | 1x0e+1x1e+1x2e | 224x0e |

*Table 10.* SevenNet-l3i5 Tensor-Product layer configuration

| | SevenNet-l3i5 | | |
|---|---|---|---|
| | **Hidden** | **Edge** | **Output** |
| **Layer 1** | 128x0e | 1x0e+1x1e+1x2e+1x3e | 128x0e+128x1e+128x2e+128x3e |
| **Layer 2** | 128x0e+64x1e+32x2e+32x3e | 1x0e+1x1e+1x2e+1x3e | 256x0e+480x1e+544x2e+480x3e |
| **Layer 3** | 128x0e+64x1e+32x2e+32x3e | 1x0e+1x1e+1x2e+1x3e | 256x0e+480x1e+544x2e+480x3e |
| **Layer 4** | 128x0e+64x1e+32x2e+32x3e | 1x0e+1x1e+1x2e+1x3e | 256x0e+480x1e+544x2e+480x3e |
| **Layer 5** | 128x0e+64x1e+32x2e+32x3e | 1x0e+1x1e+1x2e+1x3e | 256x0e |

*Table 11.* SevenNet-l4i5 Tensor-Product layer configuration

| | SevenNet-l4i5 | | |
|---|---|---|---|
| | **Hidden** | **Edge** | **Output** |
| **Layer 1** | 128x0e | 1x0e+1x1e+1x2e+1x3e+1x4e | 128x0e+128x1e+128x2e+128x3e+128x4e |
| **Layer 2** | 128x0e+128x1e+128x2e+128x3e+128x4e | 1x0e+1x1e+1x2e+1x3e+1x4e | 288x0e+576x1e+704x2e+736x3e+640x4e |
| **Layer 3** | 128x0e+128x1e+128x2e+128x3e+128x4e | 1x0e+1x1e+1x2e+1x3e+1x4e | 288x0e+576x1e+704x2e+736x3e+640x4e |
| **Layer 4** | 128x0e+128x1e+128x2e+128x3e+128x4e | 1x0e+1x1e+1x2e+1x3e+1x4e | 288x0e+576x1e+704x2e+736x3e+640x4e |
| **Layer 5** | 128x0e+128x1e+128x2e+128x3e+128x4e | 1x0e+1x1e+1x2e+1x3e+1x4e | 288x0e |

## B. Numerical Stability

We assessed the numerical stability of FlashTP against the default *e3nn* implementation as follows. First, we sampled FP32 tensors for the hidden features ($h$), edge features ($e$), and weights ($r$) from a uniform distribution over the interval (0,1). We then computed the "reference" outputs by running *e3nn* in fp64 precision. Next, we re-computed the layer outputs using both FlashTP and *e3nn* in fp32 precision. Finally, we calculated the root-mean-square error (RMSE) between each fp32 output (and its corresponding gradients) and the fp64 reference to quantify numerical stability.

Overall, results shown in Table 12 indicate that FlashTP offers improved or comparable accuracy in most cases, making it a reliable alternative.

*Table 12.* RSME of *e3nn*, and FlashTP

| Stage | Tensor | *e3nn* | FlashTP |
|---|---|---|---|
| **Forward** | out | 9.9565E-07 | 9.9171E-07 |
| **Backward** | h grad | 1.4393E-05 | 1.4393E-05 |
| **Backward** | e grad | 7.2380E-06 | 5.4978E-06 |
| **Backward** | r grad | 6.3440E-08 | 6.5808E-08 |
| **Double-Backward** | h grad | 2.5270E-07 | 3.5300E-07 |
| **Double-Backward** | e grad | 1.6240E-05 | 9.8454E-06 |
| **Double-Backward** | r grad | 1.4850E-07 | 1.3050E-07 |
| **Double-Backward** | out grad | 5.7579E-06 | 5.5367E-06 |

