# OpenReview forum: "FlashTP: Fused, Sparsity-Aware Tensor Product for Machine Learning Interatomic Potentials"
_ICML.cc/2025/Conference — ICML 2025 spotlightposter_

### Official Review · Reviewer_GZbV · 2025-02-20

**Overall Recommendation:** 3

**Summary:**

The paper presents FlashTP, a highly optimized tensor-product library designed to address computational inefficiencies through kernel fusion, sparse computation, and path-aggregated execution. The proposed approach significantly accelerates tensor-product operations in equivariant neural networks.

**Claims And Evidence:**

Overall, this paper provides clear evidence to support its claims.

In Section 3, the authors conduct a detailed empirical analysis of the computational bottlenecks in tensor-product operations. In Section 4, they explicitly explain how their proposed methods address these issues one by one. This one-to-one correspondence in problem identification and solution design, to some extent, substantiates the potential effectiveness of their approach. Furthermore, in the evaluation section, the paper presents comprehensive ablation studies, including comparisons with baseline methods in forward, backward, and double-backward propagation scenarios, as well as end-to-end experiments within neural networks.

These results collectively demonstrate that the proposed approach successfully accelerates tensor-product computations, thereby validating the claim that it enhances the efficiency of equivariant neural networks.

**Essential References Not Discussed:**

The authors should consider including a discussion on acceleration methods beyond cuEquivariance, particularly in the context of other approaches that improve efficiency.

For instance:
- Acceleration based on sparsity:
[1] Wang N, Lin C, Bronstein M, et al. "Towards Flexible, Efficient, and Effective Tensor Product Networks" (NeurIPS 2023 Workshop: New Frontiers in Graph Learning).

- Acceleration based on frequency domain:
[2] Luo S, Chen T, Krishnapriyan A S. "Enabling Efficient Equivariant Operations in the Fourier Basis via Gaunt Tensor Products" (arXiv preprint arXiv:2401.10216, 2024).
[3] Xie Y Q, Daigavane A, Kotak M, et al. "The price of freedom: Exploring tradeoffs between expressivity and computational efficiency in equivariant tensor products" (ICML 2024 Workshop on Geometry-grounded Representation Learning and Generative Modeling, 2024).

**Experimental Designs Or Analyses:**

5. In Tables 2 and 3, CuEq shows significant performance degradation when l_max > 2, even falling below e3nn. This suggests that CuEq may not have implemented adaptive modifications or optimizations for these cases. Have you checked the correctness of CuEq's execution in these cases or analyzed potential reasons for this behavior? In this context, does the comparison with CuEq still hold significant value?

6. If we focus only on the cases where l_max ≤ 2, the acceleration of FlashTP relative to CuEq seems limited, with only the Double-Backward acceleration being significantly improved. Is this behavior related to the specific design of your method?

7. According to Figure 4, assuming that tensor-product operations account for 60.3% and 75.2% of the inference and training time in SevenNet-l3i5, does this imply an upper limit on the speedup (i.e., when the tensor-product time is optimized to 0) of approximately 2.5x for inference and 4x for training? How does this conclusion reconcile with the reported speedups of "4.2× and 3.5×" for inference and training, respectively?

**Methods And Evaluation Criteria:**

Overall, the experimental design in this paper is suitable for addressing the proposed problems, though further optimization or clarification is needed in the benchmarking aspect.

The benchmarks chosen for the single tensor-product (tp) tests are e3nn and cuEquivariance, while for the end-to-end network tests, SevenNet-l3i5 is used. To my knowledge, cuEquivariance uses DiffDock and MACE for its end-to-end experiments, also demonstrating the difference between its implementation and theoretical limits. Based on this,

1. why were DiffDock and MACE not chosen or included as additional end-to-end benchmarks for a more direct comparison of performance between the two methods?

2. Does your method have any theoretical limit differences when compared to cuEquivariance?

In the experiments, the chosen node feature dimension is 32, which is common, but it shows a noticeable gap compared to the typical dimensions (128 or 256) used in other popular equivariant networks, such as the Equiformer series.

3. Does your method scale well to tensor features with a larger channel size? If so, is this expansion a simple extension, or does it require additional specialized design or implementation?

4. Is there a channel size limit based on the CUDA kernel register storage limits?

[1]Liao Y L, Smidt T. Equiformer: Equivariant graph attention transformer for 3d atomistic graphs[J]. arXiv preprint arXiv:2206.11990, 2022.

[2]Liao Y L, Wood B, Das A, et al. Equiformerv2: Improved equivariant transformer for scaling to higher-degree representations[J]. arXiv preprint arXiv:2306.12059, 2023.

[3]https://developer.nvidia.com/blog/accelerate-drug-and-material-discovery-with-new-math-library-nvidia-cuequivariance/

**Other Comments Or Suggestions:**

N/A

**Other Strengths And Weaknesses:**

Strengths:
1. The paper clearly describes the background and necessity of tensor-product acceleration, providing a detailed analysis of the speed bottlenecks in current tensor-product computations. This leads naturally to the proposed solution and evaluations, presenting a coherent story with a very smooth and readable writing style.
2. The method section offers clear motivation and solutions, and the source code is provided in the appendix, which enhances the credibility and reproducibility of the work.
3. The experimental design in the method section is comprehensive, particularly with the inclusion of the Double-Backward time tests, ensuring its applicability to MD tasks. The results also demonstrate significant speedups, confirming the potential of the method in real-world applications.

 Weaknesses:
1. The paper lacks a discussion and validation on whether the proposed method can be applied to other general equivariant models, which would significantly impact the overall quality of the paper and its real influence in the community.
2. The experimental section still contains aspects that need further clarification and optimization to strengthen the credibility of the results and to establish the speed advantages over existing methods. Further detailed discussions in other sections can serve as a reference.
3. While the method presented in the paper is effective, there seems to be limited claim about its originality and innovation. Additionally, since the paper does not have a section discussing related works, the innovations in FlashTP, as described in Chapter 4, are not directly compared to existing methods. For example, Chapter 4.3 discusses "Applying Sparsity in Tensor-Product," but the application of CG sparsity has already been discussed and applied earlier, as seen in the referenced papers mentioned above.

**Questions For Authors:**

Most of the questions have already been raised in the sections above. Additionally:

8. Do you have plans to open-source the work presented in this paper?

9. Regarding Inter-Layer Kernel Fusion, is the fusion heavily tied to the message-passing function, making its extension to other networks a non-trivial task?

Overall, I am inclined to believe that there is significant engineering work behind this paper, but I still have concerns about the reliability of the experimental results, the innovation and generalization of the method. If the authors are unable to clarify or address these issues, I may reconsider my evaluation.

**Relation To Broader Scientific Literature:**

I believe the paper has sufficiently discussed the connection between tensor product computations and various prediction tasks in scientific fields.

**Theoretical Claims:**

The paper does not contain much content regarding theoretical claims. It would be beneficial to add a discussion about the theoretical maximum speed of the proposed method, as well as a scaling analysis of the tensor product with respect to the tensor order in comparison to the baseline methods.

---

> ### Author Rebuttal · Authors · 2025-04-01
>
> Thank you for your valuable feedback.
>
> # R1. Why Diffdock/MACE was not selected for end-to-end evaluation [Q1]
> - DiffDock is a diffusion-based molecular docking model, not a MLIP model. Its CGTP configuration is different from the configurations used in MLIPs, so it falls outside the scope of FlashTP.
> - Please refer our response R2 to Reviewer LtnP for why SevenNet was chosen over MACE.
>
> # R2. Roofline analysis of FlashTP [Q2]
> - We estimate the theoretical performance limits of FlashTP using a roofline analysis based on the peak capabilities of the A100 GPU (19.5 TFLOPS and 1.9 TB/s memory bandwidth). Under this model, the estimated latency for the kernel microbenchmark with $l_{\text{max}} = 3$ is 0.46 ms for the forward pass, 0.89 ms for the backward pass, and 1.35 ms for the double-backward pass.
> - While the exact derivation of the "Speed of Light" (SoL) latencies reported in the cuEq blog [1] is not publicly documented, FlashTP’s exploitation of sparsity in the Clebsch–Gordan coefficient matrix reduces both theoretical memory traffic and floating-point operations. Consequently, the SoL latency for FlashTP is expected to be lower than that of cuEq.
>
> # R3. Channel scaling in FlashTP [Q3, Q4]
> - FlashTP supports various channel sizes without requiring any modifications, and the channel size is **not** limited by CUDA kernel register storage. The SevenNet variants used in the end-to-end evaluation employ multiple channel sizes (32, 64, and 128) for their features.
> - Below, we report the kernel microbenchmark result (in milliseconds) for $l_{\text{max}} = 3$ with channel sizes of 32, 64, and 128. These results demonstrate that the performance improvement of FlashTP remains consistent across different channel sizes.
>
> |Phase|Channel|e3nn|FlashTP (Speedup)|
> |-|-|-|-|
> |Forward|32|18.40|2.54 (7.2x)|
> ||64|34.25|5.02 (6.8x)|
> ||128|66.80|9.98 (6.7x)|
> |Backward|32|85.00|6.15 (13.8x)|
> ||64|163.96|12.14 (13.5x)|
> ||128|324.73|24.19 (13.4x)|
> |Double Backward|32|254.13|8.73 (29.1x)|
> ||64|503.28|17.41 (28.9x)|
> ||128|1003.95 |34.81 (28.8x)|
>
>
> # R4. Discussion on cuEq performance characteristics [Q5, Q6]
> - Please view our response R3 to Reviewer LtnP.
>
> # R5. Contributions regarding sparsity
> - First, it is important to clarify that the sparsity discussed in [2] refers to pruning-based sparsity in CG tensor-product (CGTP) paths, not the sparsity in the Clebsch–Gordan (CG) coefficient matrix exploited by FlashTP. These are orthogonal techniques.
> - While the idea of leveraging sparsity in CG matrices has been explored (e.g., [3]), prior attempts have not yielded practical speedups. In fact, the sparse CGTP implementation in [3] was slower than its dense counterpart. This is often due to the overhead from metadata and irregular control flow present in sparse GPU computations, which can outweigh computational gains—a well-known issue in deep learning [4].
> - In contrast, we believe FlashTP makes a significant contribution by successfully leveraging sparsity to obtain measurable performance improvements as demonstrated in our ablation study. This is enabled by introduction of efficient sparse data structure and careful kernel design that leverages constant memory.
>
> # R6. Discrepancy between Figure 4 and reported inference speedup [Q7]
> We apologize for the confusion and would like to clarify the source of the discrepancy:
> - The difference stems from the number of atoms (or more precisely, edges) used in the two settings. Inference breakdown in Figure 4 is obtained from the forward pass during training (~500 atoms), whereas the reported inference speedup is based on a MD simulation on a larger system with 4,096 atoms. In the 4K-atom setting, the tensor product accounts for approximately 88% of the total inference time, suggesting a maximum achievable speedup of around 8×. We will update Figure 4 in our final draft.
>
> # R7. Discussion about other acceleration methods
> - Please view our response R3 to Reviewer hLSR.
>
> # R8. Will FlashTP be open-sourced [Q8]
> - Yes, the code will be open-sourced. For more details on our future plans, please refer to our response R4 to Reviewer hLSR.
>
> # R9. Removing the inter-layer fusion [Q9]
> - Without the use of message-passing, inter-layer fusion becomes unnecessary. Removing this fusion requires only a minimal code change, which we have implemented for debugging purposes.
>
> # References
> [1] https://developer.nvidia.com/blog/accelerate-drug-and-material-discovery-with-new-math-library-nvidia-cuequivariance \
> [2] Wang et al., Towards Flexible, Efficient, and Effective Tensor Product Networks, NeurIPS 2023 GLFrontiers Workshop.\
> [3] Xie et al., The price of freedom: Exploring tradeoffs between expressivity and computational efficiency in equivariant tensor products, ICML 2024 Workshop GRaM.\
> [4] Wen, et al., Learning structured sparsity in deep neural networks, NeurIPS 2016.

---

### Official Review · Reviewer_LtnP · 2025-03-10

**Overall Recommendation:** 4

**Summary:**

This paper presents FlashTP, an optimized tensor-product library designed to improve the computational efficiency of equivariant machine-learning models that employ spherical tensors. The authors identify three key inefficiencies in existing tensor-product layers: excessive memory traffic from intermediate data, memory spikes from large output tensors, and lack of sparsity exploitation in the Clebsch-Gordan coefficients. FlashTP addresses these challenges through kernel fusion, sparse tensor-product computation, and path-aggregated execution. Experimental results show significant speedups over e3nn and NVIDIA cuEquivariance.

## update after rebuttal

The authors addressed all my questions. I increased my score to 4 and recommend this work for publication.

**Claims And Evidence:**

The paper claims that FlashTP significantly accelerates tensor-product operations while reducing memory usage. The experimental results support this claim, showing notable performance improvements across different benchmarks. However, the comparison with cuEquivariance is primarily numerical, lacking a discussion of key differences. It remains unclear whether FlashTP's speedup stems from algorithmic advancements or benefits from being more constrained to architectures similar to NequIP and SevenNet.

**Essential References Not Discussed:**

The paper does not mention works that explore machine-learned force fields in bases other than spherical. Given the growing body of research on Cartesian representations (see, e.g., above references), mentioning these studies could offer valuable context for the paper’s claims.

**Experimental Designs Or Analyses:**

The experiments convincingly show that FlashTP improves computational efficiency but are focused solely on models using spherical tensors. A broader comparison would strengthen the evaluation by mentioning Cartesian-based models and architectures and incorporating symmetric contraction for spherical models.

**Methods And Evaluation Criteria:**

The proposed library is well-aligned with the problem, and the evaluation consists of runtime and memory benchmarks on an NVIDIA A100 GPU. While these benchmarks provide valuable insight, the paper could benefit from additional evaluations on different MLIP architectures, particularly models incorporating symmetric contraction (e.g., MACE).

**Other Comments Or Suggestions:**

Please refer to the comments in previous sections for all comments and suggestions.

**Other Strengths And Weaknesses:**

Please refer to the comments in previous sections for all strengths and weaknesses of the presented work.

**Questions For Authors:**

My questions would relate to the issues or comments raised above, so addressing them would suffice to change my evaluation of the paper.

**Relation To Broader Scientific Literature:**

The paper addresses an important issue in the context of equivariant machine learning models. However, it focuses exclusively on models in the spherical basis and does not mention other bases, such as the Cartesian one. Including a discussion of these alternatives would offer a more comprehensive understanding of how FlashTP fits within the broader landscape of MLIP development. Some references for Cartesian models include: https://arxiv.org/abs/2306.06482, https://arxiv.org/abs/2405.14253, and https://arxiv.org/abs/2412.18263 (and references therein).

**Theoretical Claims:**

The paper suggests that higher-rank tensors (larger $l_\mathrm{max}$ values) are important to consider when improving the numerical efficiency of tensor-product layers in models designed for atomistic simulations, using this argument to support the advantages over NVIDIA cuEquivariance. However, this claim is not entirely accurate—higher-rank tensors are generally only necessary in environments with higher local symmetry, typically relaxed in atomistic simulations. A more general argument beyond atomistic simulations might provide a stronger justification.

---

> ### Author Rebuttal · Authors · 2025-04-01
>
> Thank you for your valuable feedback.
> # R1. Discussion regarding Cartesian-based models
> - Please refer to the last paragraph of our response R3 to Reviewer hLSR.
>
> # R2. Choice of SevenNet over MACE for end-to-end evaluation
> - We chose SevenNet over MACE for end-to-end evaluation because 1) we believe SevenNet more accurately reflects the current SOTA, as demonstrated by its strong performance on a widely used benchmark [1]; and 2) its architecture is more broadly representative than that of MACE. SevenNet is primarily composed of CGTP-based interactions—a design principle shared by many other leading models, such as GNoME and Allegro. In contrast, MACE introduces a unique operation called _symmetric contraction_ on top of CGTP, which is specific only to MACE and limits its generality as a benchmark.
> - Meanwhile, FlashTP can still be applied to accelerate MACE through a hybrid approach with cuEq. Specifically, the CGTP components within MACE can benefit from FlashTP, while the symmetric contraction operation—which falls outside the scope of FlashTP—can continue to be handled by cuEq. In our evaluation, this complementary use of FlashTP and cuEq reduces the per-epoch training time of the MACE-MP Large model (fp32) by 1.2× compared to using cuEq alone.
>
> # R3. Speculation on the performance differences between FlashTP and cuEq
> Since cuEq’s CUDA implementation is not open-source, identifying the exact performance bottlenecks of cuEq is infeasible. As a result, we refrained from including potentially premature speculation in the manuscript. Nonetheless, we have conducted an in-depth analysis of cuEq to the extent possible. Below, we share our informed speculations regarding the observed performance differences between FlashTP and cuEq.
>
> ## Performance degradation of cuEq for higher values of $l_{max}$.
> - cuEq launches different CUDA kernels depending on the CGTP configuration. In particular, the kernel used for CGTPs with $l_{max} \leq 2$ (cuEq-fast) differs from the one used for $l_{max} > 2$ (cuEq-slow).
>
> - Nsight Compute profiling indicates that while both cuEq-fast and cuEq-slow are L1-bandwidth-bound, their L1 utilization differs markedly: cuEq-fast achieves high L1 utilization, whereas cuEq-slow exhibits significantly lower L1 utilization. We believe this disparity is a primary contributor to the degraded performance of cuEq at higher $l_{\text{max}}$ values. Note that our references to L1 bandwidth/utilization refer to the combined bandwidth/usage of the L1 cache and shared memory, which share the same underlying hardware.
> - We speculate that the difference in L1 utilization arises from how operands are managed. Specifically, cuEq-slow shows substantially higher shared memory usage compared to cuEq-fast. Our hypothesis is that cuEq-fast handles operands primarily using registers, while cuEq-slow relies more on shared memory due to operand sizes exceeding register capacity. This increased reliance on shared memory may reduce kernel occupancy, which in turn lowers L1 utilization and ultimately impacts performance.
> ## Source of speedup for FlashTP
> - FlashTP employs a shared memory–based design regardless of the $l_{max​}$ value, yet it does not experience performance degradation due to low occupancy and consistently outperforms cuEq. This is because FlashTP substantially reduces overall memory traffic—including L1 traffic—such that low occupancy and the resulting lower L1 utilization do not become performance bottlenecks. A suite of memory traffic optimization techniques, including effective kernel fusion and path aggregation, are the key differentiator that sets FlashTP apart from cuEq in both efficiency and scalability.
>
>
> # R4. On the practical benefits of higher $l_{max}$
> - While it is true that increasing $l_{max}$​ can lead to diminishing returns in accuracy relative to the added computational cost, numerous empirical studies show that modest increases can yield meaningful improvements. For example, models such as MACE (using $l_{max}$​ from 0 to 2), SevenNet (2–3), NequIP (0–3), eSCN (6), and EquiformerV2 (6) demonstrate improved accuracy when employing higher $l_{max}$​ settings. The range of $l_{max}$​ used in our main evaluation (up to 5) is therefore well-aligned with standard practice in MLIP research and reflects a practically useful regime.
>
> # R5. Architectural details of SevenNet variants and NequIP usage
> - The tensor-product configurations used for the kernel microbenchmarks are detailed in Table 4 of Appendix A.1.
> - **NequIP** was _not_ used in the evaluation.
> - We apologize for the omission of configuration details for the three SevenNet variants. This information will be added to Appendix A.4 in the final draft.
>
> # References
> [1] Dunn et al., Benchmarking Materials Property Prediction Methods: The Matbench Test Set and Automatminer Reference Algorithm, npj Computational Materials 2020.

---

> > ### Comment · Reviewer_LtnP · 2025-04-02
> >
> > Based on the authors' response, I will raise my score from 3 to 4.

---

> > > ### Author Response · Authors · 2025-04-07
> > >
> > > Thank you for your thoughtful comments and for taking the time to consider our response. We’re glad our clarifications helped address your concerns, and we sincerely appreciate your updated score and recognition of the work’s contributions.

---

### Official Review · Reviewer_hLSR · 2025-03-12

**Overall Recommendation:** 4

**Summary:**

In this paper, the authors develop FlashTP, an optimized tensor-product library that uses kernel fusion, sparse computation, and path-aggregated execution. FlashTP achieves significant performance improvement in terms of increasing throughput and decreasing memory usage compared to common libraries, e3nn and cuEquivariance. They authors validation the speed improvements using the SevenNet-l3i5 model.

**Claims And Evidence:**

All of the claims made in the paper are directly supported by evidence. The authors make claims about the inefficiencies of current TP operations, and support these claims in Figures 4, 6 and in Table 1. All of the claims about speed-ups relative to e3nn and cuEquivariance are well supported by extensive experimental results. Additionally, the authors provide a good ablation study to demonstrate the contributions of each part of FlashTP.

**Essential References Not Discussed:**

On the MLFF side, one potential reference that is missing is [1], which is an orthogonal approach used to improve the speed of TP operations. I am not very familiar with low-level ML systems research and likely would not be aware of any missing references. However, one paper that comes to mind that maybe should have been included is FlashAttention [1], which famously used kernel fusion to accelerate LLMs.

[1] Reducing SO(3) Convolutions to SO(2) for Efficient Equivariant GNNs, Passaro et al, https://arxiv.org/abs/2302.03655
[2] FlashAttention: Fast and Memory-Efficient Exact Attention with IO-Awareness, Dao et al, https://arxiv.org/abs/2205.14135

**Experimental Designs Or Analyses:**

As mentioned in the Methods And Evaluation Criteria section, all the experiments/analysis are well designed are the models/datasets used are good representative samples of the MLFF research community as a whole. I would not expect to see noticeably different results on different models/datasets, at least in terms of the relative performance of FlashTP to e3NN/cuEquivariance.

**Methods And Evaluation Criteria:**

The two main efficiency metrics of interest to downstream researchers are throughput/latency and GPU memory usage, which are the primary metrics reported in the paper. There exist more sophisticated ways to quantify performance such as the Roofline model, and the evaluation/discussion might be better if such metrics were used, but the reality is that MLFF researchers only care about how fast the model runs and how much memory they will need to run it.

For downstream model evaluations, the authors use SevenNet-l3i5, which is a fair representative model for equivariant MLFFs as it shares the same backbone (NequIP) as many other models (MACE, GNoME, etc.)

The authors use a reasonable dataset to evaluate on (MPF). The choice of the dataset doesn't really matter and the authors include a study on scaling the number of atoms in the system,  thereby demonstrating applicability to nearly all other MLFF datasets.

**Other Comments Or Suggestions:**

### Update After Rebuttal

The authors include some additional experiments to show that the proposed method works in a more realistic multi-GPU training setup and improved the performance analysis using the Roofline model. This is a strong work the potential to have a large impact on the MLFF community.

**Other Strengths And Weaknesses:**

Strengths:
- S1: The paper is well written and provides good explanations of why existing TP libraries have huge performance bottlenecks. Figure 6 is a fantastic illustration of this.
- S2: FlashTP has very strong performance improvements over existing libraries
- S3: The experiments in the paper are well designed and representative of MLFF research/use cases.
- S4: The authors provide an initial (and minimal) codebase for FlashTP.
- S5: The paper includes a good ablation study to further analyze the creation of their fused kernels.

Weaknesses:
- W1: It is impractical to train top MLFF foundation models on a single GPU. Does FlashTP currently work on multi-GPU and can the authors provide such experiments?
- W2: This paper has the potential to be massively impactful, but only if the authors continue to work to build out their FlashTP library including writing good tutorials and adding more of the existing functionality in e3nn.
- W3: The authors evaluation of GPU performance evaluation could be more nuanced using something like the Roofline model

**Questions For Authors:**

- Q1: The main thing preventing me from giving this work a 5 is that they do not evaluate on multi-GPU training. Can the authors provide such an evaluation and demonstrate performance scaling on as many GPUs as possible?
- Q2: What is the authors plan to deploy the proposed code? There are many different operations/utilities in e3nn, do the authors plan to release a more developed software library to replace e3nn? For MLFF researchers, the ideal case would be for FlashTP kernels to be integrated into the existing e3nn codebase to maximize code reusability. In my view, this work is only a strong contribution to the community if the authors can continue to develop the software for people to use.

**Relation To Broader Scientific Literature:**

MLFF foundation models have recently become incredibly powerful and useful for a variety of downstream tasks. Most of the top performing foundation models heavily rely on TP operations, and as such are incredibly slow. Additionally, the TP operations are known to be bounded by memory latency and often have poor GPU utilization. This work takes a massive step in this direction towards speeding up these foundation models.

**Theoretical Claims:**

N/A -- no theoretical claims are made.

---

> ### Author Rebuttal · Authors · 2025-04-01
>
> Thank you for your valuable feedback.
> # R1. Evaluation on multi-GPU training [W1, Q1]
> - Multi-GPU training with FlashTP can be performed using PyTorch's Distributed Data Parallel (DDP). The table below shows the one-epoch training time (in seconds) for SevenNet-l3i5 on the MPF dataset, using varying numbers of GPU nodes. Each node consists of eight NVIDIA A100 GPUs (80 GB), interconnected with NVLink and NVSwitch, while inter-node communication is handled via a 100 GB/s network. A per-GPU batch size of 16 was used for all runs. FlashTP exhibits strong scalability across multiple GPUs. The modest reduction in speedup at higher GPU counts is likely due to increased inter-node communication overhead.
>
> | # of Nodes (# of GPUs) |1 (8)|2 (16)|4 (32)|8 (64)|
> |-|-|-|-|-|
> |e3nn|735|399|218|123|
> |FlashTP|164|87|49|30|
> |Speedup|4.5|4.6|4.4|4.1|
>
> # R2. Essential references not discussed
> - FlashAttention is a well-known work that optimizes self-attention via kernel fusion, and it served as an inspiration for the naming of FlashTP. We will include a reference to FlashAttention in Section 4.2 of the final draft.
>
> # R3. Acceleration of Clebsch-Gordan Tensor Product (CGTP)
> We appreciate the reviewers' insightful comments regarding alternative acceleration methods for CGTP. Below, we summarize several relevant approaches. We will incorporate this discussion into the final draft.
> - **SO(2) Tensor Product [1]**: This technique reduces the number of non-zero elements in the Clebsch–Gordan coefficient matrix by aligning the axis via rotation, thereby lowering computational complexity. This method is orthogonal to FlashTP and can be synergistic—FlashTP can also benefit from a reduced number of non-zero elements in the input.
> - **Gaunt Tensor Product (GTP) [2]**: GTP leverages Fast Fourier Transforms to accelerate CGTP computations by operating in the frequency domain. While computationally efficient, it trades off some expressivity compared to CGTP. Due to the symmetric nature of its operations, GTP is unable to capture chiral features in 3D structures [3].
> - **Fused Tensor Product (FTP) [4]**: FTP aggregates all irreducible representations (irreps) into a single tensor, applies native matrix multiplication, and subsequently decomposes the result to recover the original irreps. This reduces computational cost compared to CGTPs, but similar to GTP, it exhibits reduced expressivity [3]. While it has slightly higher computational complexity than GTP, it is asymmetric operations, making it more suitable for capturing parity-sensitive interactions.
> - **Cartesian-based models**: These models employ Irreducible Cartesian Tensors (ICT) instead of spherical harmonics as the basis. Recent studies have demonstrated promising performance in both accuracy and latency, particularly at lower ranks, when applied to MLIP tasks [5,6]. Nonetheless, CGTP-based models remain the dominant paradigm—top-performing entries in widely used MLIP benchmarks such as OC20 and Matbench continue to rely on CGTP, underscoring the critical need for further acceleration of CGTP [7,8].
>
> # R4. Future plans for FlashTP [W2, Q2]
> FlashTP is already deployed in production at a major tech company to deliver SOTA performance for their simulation workloads. We are also in active discussions with another major tech company to integrate FlashTP into their widely used open-source library for equivariant neural networks.
> - We plan to release FlashTP as a standalone Python package and integrate it into the e3nn codebase by implementing a variant of the `TensorProduct` class that utilizes FlashTP under the hood.
> - To facilitate reproducibility and adoption by the broader community, we will include a benchmarking script and a comprehensive integration example in the repository.
>
> # R5. Roofline model analysis [W3]
> - Please refer to our response R2 to Reviewer GZbV for the theoretical performance of FlashTP estimated from roofline analysis.
> - The roofline analysis from Nsight Compute shows that FlashTP kernels are compute-bound, whereas kernels of cuEq are L1 memory-bound. For more analysis on the differences between FlashTP and cuEq, please see our response R3 to Reviewer LtnP.
>
> # References
> [1] Passaro et al., Reducing SO(3) Convolutions to SO(2) for Efficient Equivariant GNNs, ICML 2023.\
> [2] Luo et al., Enabling Efficient Equivariant Operations in the Fourier Basis via Gaunt Tensor Products, ICLR 2024.\
> [3] Xie et al., The price of freedom: Exploring tradeoffs between expressivity and computational efficiency in equivariant tensor products, ICML 2024 Workshop GRaM.\
> [4] Unke et al., E3x: E(3)-equivariant deep learning made easy, CoRR 2024.\
> [5] Simeon et al., TensorNet: Cartesian tensor representations for efficient learning of molecular potentials, NeurIPS 2023.\
> [6] Zaverkin et al., Higher-Rank Irreducible Cartesian Tensors for Equivariant Message Passing, NeurIPS 2024.\
> [7] https://opencatalystproject.org/leaderboard.html \
> [8] https://matbench-discovery.materialsproject.org

---

> > ### Comment · Reviewer_hLSR · 2025-04-01
> >
> > I appreciate the response from the authors. The results of the multi-GPU and roofline analyses are strong and I highly recommend the authors include both of them in the revised manuscript, ideally in the main text.
> >
> > I will keep my score at a 4 only because I don't think this work has broad enough impact to be an oral paper, but I recommend that this work be accepted as a spotlight paper.

---

> > > ### Author Response · Authors · 2025-04-07
> > >
> > > Thank you for your insightful comments and encouraging feedback. As suggested, we will incorporate both the multi-GPU and roofline analyses into the revised manuscript. We also sincerely appreciate your recognition of our work and your recommendation for it to be considered as a spotlight paper.

---

### Decision · Program_Chairs · 2025-05-01

**Decision:**

Accept (spotlight poster)

**Comment:**

All reviewers agree that this is an important engineering contribution in the domain of machine learning for molecular applications.  The analogy to flash attention for transformers appears fair, and this work has the potential to inspire follow-up work or re-use in adjacent domains.